# Distinct Changes in Placental Ceramide Metabolism Characterize Type 1 and 2 Diabetic Pregnancies with Fetal Macrosomia or Preeclampsia

**DOI:** 10.3390/biomedicines11030932

**Published:** 2023-03-17

**Authors:** Miira M. Klemetti, Sruthi Alahari, Martin Post, Isabella Caniggia

**Affiliations:** 1Lunenfeld-Tanenbaum Research Institute, Sinai Health System, Toronto, ON M5T 3H7, Canada; 2Obstetrics and Gynecology, University of Helsinki and Helsinki University Hospital, 00029 HUS Helsinki, Finland; 3Program in Translational Medicine, Peter Gilgan Centre for Research and Learning, Hospital for Sick Children, Toronto, ON M5G 0A4, Canada; 4Institute of Medical Science, University of Toronto, Toronto, ON M5S 1A1, Canada; 5Department of Physiology, University of Toronto, Toronto, ON M5S 1A1, Canada; 6Department of Obstetrics & Gynecology, University of Toronto, Toronto, ON M5S 1A1, Canada

**Keywords:** ceramide, placenta, diabetes, preeclampsia, macrosomia, sphingosine kinase

## Abstract

Disturbances of lipid metabolism are typical in diabetes. Our objective was to characterize and compare placental sphingolipid metabolism in type 1 (T1D) and 2 (T2D) diabetic pregnancies and in non-diabetic controls. Placental samples from T1D, T2D, and control pregnancies were processed for sphingolipid analysis using tandem mass spectrometry. Western blotting, enzyme activity, and immunofluorescence analyses were used to study sphingolipid regulatory enzymes. Placental ceramide levels were lower in T1D and T2D compared to controls, which was associated with an upregulation of the ceramide degrading enzyme acid ceramidase (ASAH1). Increased placental ceramide content was found in T1D complicated by preeclampsia. Similarly, elevated ceramides were observed in T1D and T2D pregnancies with poor glycemic control. The protein levels and activity of sphingosine kinases (SPHK) that produce sphingoid-1-phosphates (S1P) were highest in T2D. Furthermore, SPHK levels were upregulated in T1D and T2D pregnancies with fetal macrosomia. In vitro experiments using trophoblastic JEG3 cells demonstrated increased SPHK expression and activity following glucose and insulin treatments. Specific changes in the placental sphingolipidome characterize T1D and T2D placentae depending on the type of diabetes and feto-maternal complications. Increased exposure to insulin and glucose is a plausible contributor to the upregulation of the SPHK-S1P-axis in diabetic placentae.

## 1. Introduction

Proper placental function is a vital determinant of maternal adaptation to pregnancy, fetal development and growth, and long-term health outcomes in both the mother and child [1,2]. Vascular and metabolic complications, such as hypertensive disorders and abnormal fetal growth, are common in pregnancies affected by type 1 (T1D) or type 2 diabetes (T2D), and the placenta is known to be centrally involved in their pathogenesis [3,4,5]. Complex disturbances of the lipidome of highly metabolic tissues and plasma are typical in diabetes [6,7,8]; however, our understanding of the changes in lipid metabolism that occur in the placenta due to maternal diabetes is incomplete.

Sphingolipids are bioactive lipids that, apart from their function as essential building blocks of the cell membrane, are critically involved in a plethora of cell signaling pathways. For example, specific sphingolipid species are implicated in cell fate, inflammatory, immune and stress responses, cell adhesion and migration, angiogenesis, vascular function, and mitochondrial bioenergetics [9,10]. In addition, they are also important regulators of cell membrane dynamics and vesicular trafficking [9]. The precursor of all sphingolipids is ceramide, the “hub” of complex sphingolipid metabolism, which is produced and degraded in a compartmentalized fashion by a multitude of enzyme variants and interconnected pathways [11]. Briefly, the de novo synthesis pathway of ceramide begins in the endoplasmic reticulum (ER) with the production of sphinganine from palmitoyl-CoA and serine in a rate-limiting reaction catalyzed by serine palmitoyltranferase. Subsequently, acyl chains of variable length (14 to 34 carbon atoms) are added to the sphingoid backbone by specific ceramide synthases (CerS1-6). Alternatively, sphinganine can be phosphorylated into sphinganine-1-phosphate (Sa1P) by sphingosine kinases (SPHK) [12]. Ceramide can also be produced by hydrolysis of sphingomyelin catalyzed by sphingomyelinases (hydrolysis pathway), or via re-acylation from sphingosine (salvage pathway) [11]. Ceramide biogenesis by these pathways can be triggered in different intracellular compartments by various stressors, such as oxidative insults, hypoxia, inflammation, or an overload of saturated fatty acids [13]. Ceramide is hydrolyzed into sphingosine (SPH) and fatty acids by ceramidases. SPH can then be phosphorylated by SPHKs to form the lysosphingolipid sphingosine-1-phosphate (S1P). Both ceramide and S1P govern various signaling pathways related to cell death, proliferation, migration, survival, and senescence [10,14].

Numerous studies have reported that disturbances of sphingolipid homeostasis are key to the onset and progression of diabetes, obesity, and associated cardiovascular conditions [8,13]. We have shown that preeclampsia, a life-threatening pregnancy hypertensive syndrome typified by maternal systemic endothelial dysfunction, is associated with placental lysosomal ceramide build-up, characterizing preeclampsia as a sphingolipid storage disorder [15]. Furthermore, we have reported that placental ceramide accumulation in preeclampsia results in increased trophoblast autophagy and necroptosis [15,16], and tilts the mitochondrial dynamic balance towards fission, leading to increased mitophagy [17]. In contrast, in fetal growth restriction, placental ceramide levels are decreased due to the upregulation of acid ceramidase (ASAH1). In addition, SPHK expression and activity are reduced, suggesting that the aberrant placental ceramide/sphingoid-1-phosphate axis might contribute to abnormal fetal growth [18]. Interestingly, we recently demonstrated a similar reduction in ceramides, in conjunction with upregulated mitochondrial fusion, in placentae affected by gestational diabetes (GDM) [19]. Collectively, these observations suggest that specific alterations in the placental sphingolipidome distinguish vasculo-metabolic complications of pregnancy and are interlinked with major alterations in placental cell metabolism and bioenergetics.

In the context of T1D and T2D pregnancies, the significance of sphingolipids has received less attention. Herein, we examined placental ceramide and sphingoid-1-phosphate content and metabolism in women with T1D and T2D. Additionally, we investigated whether common complications that burden mothers with diabetes (preeclampsia) and their fetuses (macrosomia/large-for-gestational age) are characterized by distinct placental sphingolipid profiles that could also shed light on the placental metabolic derangements implicated in the pathogenesis of these complex outcomes.

## 2. Materials and Methods

### 2.1. Collection of Placental Samples and Clinical Information

Pregnant women with T1D (*n* = 28) and T2D (*n* = 21) were recruited by the Research Centre for Women’s and Infants’ Health (RCWIH) Biobank, Mount Sinai Hospital (MSH), Toronto, Canada. Healthy women without diabetes and with a delivery between 34+0- and 40+5-weeks’ gestation (*n* = 49) were recruited as controls. Only singleton pregnancies were included in the study. Smokers (self-reported, any time of gestation) and pregnancies affected by substance abuse, fetal malformations, chromosomal aberrations, chorioamnionitis, or small-for-gestational age (relative birth weight <−2.0 SD units) in the absence of preeclampsia were excluded. All participants provided written informed consent. Placental tissue was collected according to the ethical guidelines of the University of Toronto, Faculty of Medicine and MSH, immediately after delivery, and snap-frozen in liquid nitrogen. The study protocol was approved by the MSH Research Ethics Board (REB number: 11-0287-E) and carried out in agreement with the Declaration of Helsinki.

Available information on maternal age, parity and gravidity, pre-pregnancy weight and height, gestational weight gain (GWG), age at diabetes diagnosis, blood pressure, glycated hemoglobin (HbA_1c_), and obstetric and perinatal outcomes were extracted from the MSH patient records. Obesity was defined as maternal pre-pregnancy body mass index (BMI) ≥ 30 kg/m^2^. GWG was categorized according to the Institute of Medicine (IOM) criteria as adequate, below, or above recommendations [20]. Preeclampsia was defined according to the criteria of the American College of Obstetricians and Gynecologists [21]. Fetal macrosomia (i.e., large-for-gestational age (LGA)) was defined as a relative birth weight (birth weight z-score) exceeding + 2 SD units (>97.7th percentile) using a Canadian standard population, standardized for sex and gestational age [22].

### 2.2. Lipid Mass Spectral Analysis

Placental tissue from 42 controls, 26 women with T1D, and 21 women with T2D were processed for lipid analysis as described previously [17]. Following lipid extraction, sphingolipid species were quantified utilizing high-performance liquid chromatography coupled to tandem mass spectrometry (LC-MS/MS) at the Analytical Facility for Bioactive Molecules, Hospital for Sick Children, Toronto, ON, Canada.

### 2.3. Western Blotting

Approximately 100–200 mg of snap-frozen placentae were crushed, dissolved in 400–600 μL of RIPA buffer containing proteasome inhibitor, and homogenized at 4 °C to generate placentae tissue lysates that were centrifuged at 12,000 rpm for 10 min. Protein concentration was quantified using Bradford protein assay (Bio-Rad^®^, Mississauga, ON, Canada). Placental lysates containing 30 μg of protein were subjected to SDS-PAGE on 7.5% (SPHK1 protein analysis), 10% (SPHK2 protein analysis), or 12% (ASAH1 protein analysis) BioRad FastCast acrylamide gels (Bio-Rad^®^, Mississauga, ON, Canada). After each run, the gels were imaged for a stain-free profile of total protein using the BioRad Chemidoc XRS+ System. This stain-free imaging technology is based on the detection of a fluorescence signal generated when trihalo compounds in the polyacrylamide gels modify tryptophan residues on exposure to ultraviolet light [23]. Subsequently, proteins were transferred onto polyvinylidene fluoride (PVDF) membranes using Trans-Blot Turbo transfer buffer 5×. Membranes were blocked with 5% (*w*/*v*) non-fat milk dissolved in Tris-buffered saline containing 0.1% (*v*/*v*) Tween 20 (TBS-T) for 1 h, before probing with respective antibodies diluted in 5% (*w*/*v*) non-fat milk at 4 °C overnight. Next day, membranes were washed 3 × 10 min in TBS-T, incubated with appropriate secondary antibody in 5% (*w*/*v*) non-fat milk for 1 h at room temperature, and washed 3 × 10 min in TBS-T. Finally, membranes were simultaneously visualized for immunoreactivity following the addition of chemiluminescence ECL reagent (Bio-Rad^®^, Mississauga, ON, Canada) and equal exposure time using Bio-Rad Chemidoc XRS+ System. Densitometric analysis was performed using ImageLab software, with data normalized to total protein in stain-free gels. Total protein normalization was reported to be the most appropriate normalization method [23], as diabetes and preeclampsia have both been demonstrated to affect the placental expression of various proteins that are commonly used as loading controls/housekeeping proteins [24].

### 2.4. Immunofluorescence Analysis

Immunofluorescence staining for SPHK1 and SPHK2 as well as calreticulin (CRT; ER marker), TOM20 (mitochondria marker), and zonula occludens-1 (ZO-1; plasma membrane marker) in JEG3 cells treated with 25 mM of glucose (GLU) and 0.85 μM of insulin solution, or with EMEM vehicle (VEH) alone, was performed as previously described [17]. IF intensity and co-localization (by Pearson’s correlation coefficient) were quantified using ImageJ as reported [25].

### 2.5. Antibodies

Rabbit anti-ASAH1 was from Aviva Systems Biology (Cedarlane, Burlington, ON, Canada (OAPB00726; WB 1:750). Mouse monoclonal anti-SPHK1 (SC-365401; IF 1:200), goat anti-SPHK2 (SC-22704; IF 1:200, WB 1:500), rabbit anti-ZO-1 (SC-10804; IF 1:200), and rabbit anti-TOM20 (SC-11415; IF 1:200) were purchased from Santa Cruz Biotechnology (Mississauga, ON, Canada). Rabbit monoclonal anti-SPHK1 (ab109522; WB 1:1000) and mouse monoclonal anti-calreticulin (ab22683; IF 1:300) were from Abcam (Cedarlane, Burlington, ON, Canada). Secondary antibodies used were goat anti-rabbit IgG-HRP (sc-2054; WB 1:2000), goat anti-mouse IgG-HRP (sc-2005; WB 1:2000), and donkey anti-goat IgG (H+L) (WB 1:2000) from Jackson Laboratory, Bar Harbor, Maine, USA. For IF experiments, Alexa Fluor 488 donkey anti-rabbit IgG (A21206) and Alexa Fluor 594 donkey anti-mouse IgG (A21203) were obtained from ThermoFisher Scientific (Mississauga, ON, Canada).

### 2.6. SPHK Activity Assay

Total SPHK enzyme activity (SPHK1+SPHK2) was measured in placental and JEG3 lysates using an SPHK Assay Kit (Echelon Bioscience, K-3500, Salt Lake City, UT, USA) as previously described [18].

### 2.7. RNA Analysis

Total RNA was extracted from 30 mg of frozen placental tissue using a QIAGEN RNeasy Plus Mini Kit, according to the manufacturer’s protocol. RNA quantity and purity were analyzed using a Thermo Scientific Nanodrop 1000. Next, 1 µg of RNA was reverse transcribed using the Quantabio qScript cDNA SuperMix kit (VWR International, Mississauga, ON, Canada). cDNA was then added to Quantabio PerfeCTa FastMix II and TaqMan human primers and probes for ASAH1, SPHK1, and RPLPO (Applied Biosystems, ThermoFisher Scientific, Waltham, MA, USA). qPCR was performed in duplicate on a Bio-Rad CFX96 Real-Time System, and Ct values were obtained in the Bio-Rad CFX Manager 3.1 program using regression as the Ct determination mode. Average Ct values were normalized against average expression of RPLPO (ΔCt values), then fold change between groups was analyzed using the 2^−ΔΔCt^ calculation [26]. Samples with a Ct value > 0.5 between duplicates for a certain probe were reanalyzed.

### 2.8. Cell Line Culture and Treatments

Choriocarcinoma JEG3 cells (ATCC^®^ HTB-36™, ATCC), authenticated by short tandem repeat genotyping, were cultured on coverslips in 6-well plates in standard conditions (ambient air) at 37 °C in Eagle’s Minimum Essential Medium (EMEM) (ATCC, 30–2003) containing 10% (*v*/*v*) FBS and penicillin–streptomycin (Wisent Inc., St Bruno, QC, Canada). Upon reaching 60%–80% confluency, cells were treated with either 25 mM of glucose (GLU), 0.85 μM of insulin solution (INS; 10 mg/mL in 25 mM HEPES, Catalog No. I9278, Sigma-Aldrich, Munich, Germany), glucose plus insulin (GLU+INS), or EMEM alone (VEH) for 24 h. After treatment, the cells were washed with phosphate-buffered saline (PBS) on ice and either collected at 4 °C in 1μM phenylmethylsulfonyl fluoride (PMSF) for analysis of SPHK enzyme activity and snap-frozen at −80 °C or fixed with 3.7% (*v*/*v*) formaldehyde for immunofluorescence (IF) microscopy.

### 2.9. Statistics

Statistical analyses were performed using GraphPad Prism 8 for Windows 64-bit (Version 8.1.2 (332), 6 May 2019) and IBM^®^ SPSS^®^ Statistics Version 25.0. Categorical variables were analyzed with the Chi-square test. For the comparison of continuous variables in two groups, Student’s *t* test, or Mann-Whitney U-test was utilized, as appropriate. In the case of hypotheses involving more than two comparison groups, One-way Analysis of Variance or Kruskal—Wallis tests were applied followed by Tukey’s or Dunn’s post-hoc tests, respectively. Data are presented as means (SD) or medians (IQR or range). Values of *p* < 0.05 were considered statistically significant.

## 3. Results

### 3.1. Maternal and Perinatal Characteristics

Maternal characteristics and perinatal outcomes of pregnant individuals with T1D and T2D and control pregnancies are displayed in Table 1. Maternal age and parity were similar in all groups. Age at diabetes diagnosis was lower and diabetes duration longer in women with T1D compared to those with T2D. Maternal pre-pregnancy BMI was higher in T2D pregnancies than in control women. Most individuals in all study groups gained more weight during pregnancy than recommended by the IOM. As expected, blood pressure levels after 20 weeks’ gestation and the frequencies of hypertensive complications, such as preeclampsia, were higher in women with diabetes. Of the T1D pregnancies with preeclampsia (T1DPE), two were early-onset syndromes occurring <34 weeks’ gestation and four were late-onset cases appearing ≥34 weeks’ gestation. In contrast, among the T2D women with preeclampsia (T2DPE), all patients had the early-onset subtype. Glycated hemoglobin (HbA_1c_) levels were similar in women with T1D and T2D. Most women were delivered by cesarean section without preceding labor in all groups. However, the median gestational age at birth was slightly lower in women with diabetes. Both the mean relative birth weight of the offspring and the frequency of fetal macrosomia (i.e., large-for-gestational age (LGA); birth weight z-score > +2.0 SD units) were higher in pregnancies affected by diabetes, whereas no macrosomic offspring were born to control women. In T1D pregnancies with LGA (T1DLGA), the average (range) relative birth weight of the fetuses was +2.95 (2.32–4.39) SD units, and in T2D pregnancies with LGA (T2DLGA), the average (range) relative birth weight was +3.29 (2.91–4.47) SD units.

### 3.2. Pre-Gestational Diabetes Is Associated with Decreased Placental Ceramide

Considering our previous studies showing that pregnancy disorders are associated with changes in placental sphingolipid content [15,18,19,27], we first examined placental sphingolipids in T1D and T2D pregnancies. Since LC-MS/MS analysis revealed an overall lower total ceramide content in both T1D and T2D placentae relative to controls (Table 2), we pooled the data from all diabetes cases and observed a significant decrease in total placental ceramide concentrations in diabetes (T1D+T2D: diabetes mellitus (DM)) vs. control pregnancies (Figure 1A). Correspondingly, placental concentrations of sphingosine, a bioactive product of ceramide hydrolysis by ASAH1, displayed a trend toward increased levels, although the difference was not significant (Figure 1B). With respect to distinct ceramide species (14–24 carbon acyl chains), both T1D and T2D placentae were characterized by significantly lower concentrations of CER 18:0, 22:0, and 24:1 as compared to those from control pregnancies (Figure 1C). Despite lower ceramide levels in diabetic placentae, no changes in placental sphinganine or dihydroceramides, intermediates of the de novo pathway of ceramide synthesis, were observed between control vs. diabetic pregnancies (Appendix A), suggesting that de novo ceramide pathway was unaffected by diabetes.

### 3.3. Preeclampsia and Fetal Macrosomia Result in Distinct Alterations of the Placental Sphingolipidome in Type 1 and Type 2 Diabetic Pregnancies

Placental concentrations of various ceramide species in T1D, T1DPE, and control pregnancies were next examined. No changes in ceramide species were found in placentae from T1D relative to CTR, except for the CER 24:1 concentration that was significantly lower in T1D placentae (Table 2, upper panel). In line with our earlier observations in preeclamptic pregnancies [15], total placental ceramide content as well as CER 22:0 and 24:0 species concentrations were increased in T1DPE, compared to T1D. CER 18:0 also showed elevated levels in T1DPE, when compared with T1D and CTR (*p* = 0.040), but pairwise post-hoc analyses yielded *p* > 0.05 (Table 2, upper panel). Furthermore, we found that when T1D pregnancies with a large-for-gestational age fetus (T1DLGA) pregnancies were excluded, also the median (IQR) CER 16:0 concentration was higher in T1DPE [9.73 (5.38/9.24); *n* = 6] as compared to T1D placentae [6.48 (5.56/7.06); *n* = 13] (*p* = 0.002). In contrast to T1DPE, placental levels of long- and very-long-chain ceramide species, including CER 18:0, CER 24:0, and CER 24:1, were significantly lower in T2DPE than in controls (Table 2, lower panel). Furthermore, when only preeclamptic diabetes patients were compared, lower total placental ceramide content (median (IQR)) was observed in T2DPE (35.60 (18.90/45.43)) vs. T1DPE (66.99 (58.87/71.28)) (*p* = 0.017).

Since we have previously reported that pregnancies with fetal growth restriction exhibit reduced placental CER levels [18], we next explored whether placental ceramide concentrations are affected by fetal macrosomia. T1DPE and T2DPE placentae were excluded from the analysis. LC-MS/MS analyses revealed increased placental CER 14:0 and CER 16:0 content in T1DLGA pregnancies as compared to controls and T1D, respectively (Table 3). In T2D, no differences were seen in placental ceramide levels between T2DLGA pregnancies, T2D pregnancies without macrosomia, and controls (Table 3).

### 3.4. Placental Acid Ceramidase Is Upregulated in Type 1 Diabetes and in Type 2 Diabetes with and without Preeclampsia

In line with our observation of lower placental ceramide levels in both types of pre-gestational diabetes, immunoblotting for ASAH1, the enzyme that breaks down ceramide to sphingosine, revealed significantly greater levels of this enzyme in both T1D and T2D (Figure 1D), compared to control pregnancies. ASAH1 levels varied markedly in the individual T1DPE samples, but on average trended higher; however, no statistical changes in ASAH1 protein levels were found in placental lysates from T1DPE patients relative to controls or T1D (Figure 1D). In contrast, ASAH1 protein content was upregulated in T2DPE placentae relative to controls (Figure 1D), in keeping with lower levels of very-long-chain ceramides in T2DPE. No changes in placental ASAH1 levels were observed with respect to fetal macrosomia (mean (SD) protein fold changes in CTR vs. T1DLGA: 1.00 (0.20 vs. 1.14 (0.45), *p* = 0.54; in T1D vs. T1DLGA: 1.41 (0.44) vs. 1.14 (0.45), *p* = 0.19; in CTR vs. T2DLGA: 1.00 (0.17) vs. 1.24 (0.41), *p* = 0.17; and in T2D vs. T2DLGA: 1.39 (0.37) vs. 1.24 (0.41), *p* = 0.55). Real-time PCR analysis demonstrated no changes in *ASAH1* mRNA expression in diabetes vs. control placentae (Appendix A), suggesting that the increases in ASAH1 protein in diabetes are likely due to a reduced turnover of the enzyme.

### 3.5. Poor Maternal Glycemic Control Is Associated with Elevated Levels of Placental Ceramides

Due to the observed differences in placental ceramide levels in diabetic pregnancies with and without preeclampsia and fetal macrosomia, we next explored the associations between common clinical risk factors of these disorders—obesity, gestational weight gain, and glycemic control—and placental ceramide. Since excess adiposity is known to associate with alterations of sphingolipid metabolism [28], we compared placental sphingolipids in non-diabetic obese (BMI 32.1–64.1 kg/m^2^) vs. non-obese control women (17.2–27.7 kg/m^2^). No differences in specific ceramide species nor in total placental ceramide concentrations were found in placentae from obese vs. non-obese normoglycemic women (Appendix A). Similarly, regarding gestational weight gain, we did not find placental ceramide differences between diabetic women who gained more or less than the IOM-recommended amount of weight during pregnancy. Finally, we examined sphingolipid changes in placentae from diabetic women with poor vs. better glycemic control in late pregnancy as identified by third-trimester HbA_1c_ concentrations (HbA_1c_ ≥ 64 mmol/L (≥7.5%) vs. <64 mmol/L (<7.5%)). Interestingly, we observed increased placental CER 16:0, 18:0, and 24:0 species in diabetic pregnancies with poor late-pregnancy glycemic control, which accounted for a significant increase in total placental ceramides in these pregnancies (Table 4).

### 3.6. Placental Production of Sphingoid-1-Phosphates Is Upregulated in Diabetic Pregnancies Complicated by Preeclampsia and Fetal Macrosomia

Despite their low tissue concentrations, sphingoid-1-phosphates, namely sphingosine-1-phosphate (S1P) and sphinganine-1-phosphate (Sa1P), are highly bioactive lysosphingolipids that play an integral part in the sphingolipid cycle. Hence, we extended our analyses into the quantification of these lipid species in our placental samples using high-performance LC-MS/MS. No changes in total sphingoid-1-phosphates (S1P+Sa1P) were found in T1D or T2D placentae relative to controls (Table 2). However, when preeclampsia cases were excluded from the analysis, higher median (IQR) levels of sphingoid-1-phosphates were observed in T2D (0.112 (0.054/0.282)) vs. T1D (0.087 (0.079/0.125)) placentae (*p* = 0.041). In T1DPE, on the other hand, heightened total sphingoid-1-phosphate levels were seen as compared to T1D (Table 2). T2DPE placentae displayed a similar trend towards increased sphingoid-1-phosphate levels as compared to T2D (Table 2). Combined analysis showed increased sphingoid-1-phosphates in both T1DPE+T2DPE (0.270 (0.167/0.494)) vs. T1D+T2D (0.089 (0.060/0.250), *p* = 0.041) and vs. controls (0.093 (0.067/0.138), *p* = 0.026). Likewise, examination of sphingoid-1-phosphate concentrations revealed an enrichment of sphingoid-1-phosphates in T1DLGA placentae compared to controls or T1D (Table 3). No changes in sphingoid-1-phosphates content were seen with respect to LGA in T2D placentae.

### 3.7. Placental Sphingosine Kinase Activity Is Increased in Type 2 Diabetic Pregnancies and in Diabetic Pregnancies with Fetal Macrosomia

Considering our findings of decreased placental ceramides in both T1D and T2D, and increased placental sphingoid-1-phosphate levels in T2D and both types of diabetes with LGA, we next examined the concentrations and activity of sphingosine kinase isoenzymes (SPHK1/2), which have the potential to turn the placental sphingolipid rheostat towards S1P production (Appendix A). Western blotting showed increased placental SPHKs (SPHK1+SPHK2) concentrations in T2D, but not T1D, and in both T1DLGA, and T2DLGA relative to controls (Figure 2A). Since SPHK isoforms elicit partly overlapping and compensatory roles [29,30], we next quantified total SPHK levels in diabetic and non-diabetic placental tissue. Densitometric analyses confirmed a significant increase in SPHKs in T1DLGA and T2DLGA placentae compared to controls and in T2DLGA relative to T2D (Figure 2B). Despite increased sphingoid-1-phosphate levels in diabetic pregnancies with preeclampsia, no changes were seen in placental SPHK1 or SPHK2 enzyme levels in either diabetes type in the presence of preeclampsia. In line with increased placental SPHK protein levels and total sphingoid-1-phosphates in T2D, SPHK activity was significantly increased in T2D placentae, but not in T1D, as compared to controls (Figure 2C). Again, no differences in *SPHK1* mRNA expression were observed between control, T1D, T1DLGA, T2D, or T2DLGA placentae (Appendix A). Thus, like ASAH1, the increase in SPHK1 protein in T2D and T1DLGA/T2DLGA placentae is likely due to a slower turnover of the enzyme.

### 3.8. In Vitro Exposure of Trophoblast Cells to High Glucose and Insulin Augments SPHK Activity

To assess the effect of diabetic milieu on SPHK activity in vitro, we treated choriocarcinoma JEG-3 cells with 0.85 μM insulin (INS), 25 mM glucose (GLU), INS+GLU, or control vehicle EMEM (VEH) for 24 h. In line with the observations in T2D placentae, SPHK activity was markedly increased in cells treated with GLU or GLU+INS (Figure 3A). SPHK1 has been reported to primarily reside in the cytosol and plasma membrane, while SPHK2 is largely restricted to the mitochondria, ER, and nuclear compartments [31]. Consistent with increased enzyme activity, IF revealed an enhanced positive signal of both SPHK1 and SPHK2 in JEG3 cells following GLU+INS exposure (Figure 3B). SPHK1 signal co-localization with the plasma membrane marker ZO-1 was increased following GLU+INS treatment (Pearson’s correlation coefficient of 0.47 in GLU+INS treatment vs. 0.26 in vehicle-treated controls; Figure 3B), indicating a translocation to the plasma membrane following enzyme activation. Similarly, SPHK2 showed markedly enhanced co-localization with the mitochondrial marker TOM20 (Pearson’s correlation coefficient of 0.39 in GLU+INS treatment vs. 0.28 in Vehicle-treated controls; Figure 3C) and the ER marker calreticulin (Pearson’s correlation coefficient of 0.3350 in GLU+INS treatment vs. 0.2386 in vehicle-treated controls; Figure 3D). Taken together, these data indicate augmented expression and activation of SPHK enzymes in placental cells upon in vitro exposure to hyperglycemic and hyperinsulinemic conditions.

## 4. Discussion

In the present study, we provide evidence of changes in placental ceramide metabolism in pregnancies complicated by pre-existing maternal diabetes. We found that T1D and T2D pregnancies are characterized by reduced placental ceramide content; however, in the presence of preeclampsia, T1D pregnancies exhibit placental ceramide enrichment. Of clinical relevance, we report that poor glycemic control, an important risk factor of fetal overgrowth and other adverse pregnancy outcomes [4], is associated with increased placental ceramides. In agreement with this, elevated levels of pro-apoptotic CER 16:0 were also observed in placentae from T1D pregnancies with a macrosomic fetus. Moreover, we show that placental SPHKs are upregulated in T2D and in both types of diabetes with fetal macrosomia, in line with augmented levels of sphingoid-1-phosphates. Finally, our in vitro experiments demonstrate that hyperglycemic and hyperinsulinemic conditions, typical of T1D and T2D, are plausible contributors to the upregulation of SPHK activity in diabetic placentae.

To the best of our knowledge, our study provides the first examination of placental ceramide and lysosphingolipid metabolism in the context of pregnancies affected by T1D and T2D. Proper sphingolipid metabolism and signaling are vital for embryo implantation, organogenesis, and fetal growth [30,32,33,34,35,36]. In well-controlled GDM, we have recently reported decreased placental CER 16:0, 18:0, and 24:0 levels in conjunction with an upregulation of the ASAH1 enzyme [19]. Similarly, a recent study, using high-resolution mass spectrometry, showed lower total ceramides, and particularly reduced levels of CER 14:0, 16:0, and 18:0 species, in placentae from GDM vs. non-GDM pregnancies [37]. In contrast, another study using immunohistochemical analysis suggested increased ceramides in placental villous trophoblasts of insulin-treated GDM patients [38]. Our present findings on decreased placental ceramides in T1D and T2D and these published observations in GDM placentae collectively suggest that changes in this core sphingolipid typify diabetic placental metabolism. In the present study, placental ASAH1 upregulation was most prominent in T2D, which shares pathogenesis with GDM and obesity. However, our analysis of placental sphingolipid profiles in obese vs. non-obese control pregnancies suggested that the reduction in placental ceramides is not due to obesity but rather caused by mechanisms related to diabetes. This is in accordance with a previous report indicating that GDM is associated with a decrease in placental ceramides irrespective of maternal BMI [37]. The metabolic features of T2D are particularly likely to “fuel” the production of ceramide [39], and active compensatory mechanisms are needed to optimize feto-placental well-being. Interestingly, outside pregnancy, in other metabolically active tissues, the overexpression of ASAH1 [40] and SPHK [41] has been demonstrated to prevent ceramide accretion, leading to improved glucose and lipid metabolism. Considering that in GDM reduced placental ceramides were associated with the dominance of mitochondrial fusion [19], our current results warrant further investigations into the relevance of ceramide in placental mitochondrial function and dynamics in pre-gestational diabetes.

Emerging evidence has elucidated the diverse roles of different sphingolipid species, which are dependent on the length of their carbon chain and intracellular localization, in human metabolism and its disorders [42,43]. Although tissue-specific exceptions exist, CER 16:0 and 18:0 species are generally pro-apoptotic, CER 16:0, 18:0, and 22:0 anti-proliferative, and CER 14:0 and 16:0 pro-autophagic in different cellular processes [43]. Our data indicate that these species were increased in the placentae of T1D women with preeclampsia. This is in line with our previous reports in early-onset preeclampsia, featuring CER-dependent increased trophoblast cell death, autophagy, and necroptosis rates [15,16]. Placental oxidative stress, which burdens both preeclamptic and diabetic placentae [44], likely promotes ceramide accumulation, as we have previously shown that the induction of oxidative stress in placental villous explants and JEG-3 cells upregulates CER 16:0 and 18:0 [15]. Since severe hyperglycemia in T1D increases the likelihood of oxidative stress [45], it could contribute to increased CER 16:0, 18:0, and 24:0 levels in women with poor late-pregnancy glycemic control. In contrast to our finding on T1DPE, placental ceramide enrichment did not characterize T2DPE, but, instead, lower levels of very-long-chain ceramides (CER 24:0 and 24:1) were found. This could be due to the upregulation of ASAH1 and SPHK which reduce ceramide and sphingosine levels, respectively. Even in non-diabetic pregnancies, preeclampsia is characterized by hyperinsulinemia, and other features of the metabolic syndrome [46], and it is plausible that these metabolic characteristics in T2DPE could contribute to ASAH1 [19] and SPHK [29] upregulation. On the other hand, since CER 24:0 and 24:1 have been reported to have pro-proliferative effects in some tissues, their reduced levels could also have opposite effects [9,43]. Decreased plasma concentrations of very-long-chain ceramides have also been implicated in diabetic nephropathy [47], which shares risk factors and pathophysiological background with preeclampsia, as both disorders feature endothelial dysfunction and glomerular damage leading to hypertension and proteinuria.

Sphingosine kinases (SPHKs) are key regulators of sphingolipid metabolism, as they are responsible for the phosphorylation of sphingosine and sphinganine into potent signaling lysosphingolipids, S1P and Sa1P, respectively. S1P is a powerful mediator of cell proliferation and survival, vascularization, and angiogenesis, with many effects opposed to those of ceramide [14,48]. The functions of Sa1P are less clear, but mouse studies have demonstrated tissue-protective effects in ischemia and reperfusion injury of the liver [49]. SPHK1 and SPHK2 subcellular distribution contribute to the diverse functions [10] and compartmentalization of sphingoid-1-phosphate production [14,50]. Interestingly, while modest increases in SPHK2 levels, like those observed in the present study in diabetic placentae, have been reported to promote cell proliferation and survival, high levels of this enzyme may lead to pro-apoptotic signaling [50,51]. Congruent with our IF data in trophoblast cells, in other tissues, SPHK1 has been reported to locate in the cytosol from where it shuttles to the plasma membrane to elicit its effects, whereas SPHK2 resides mostly in the mitochondria, ER, and nucleus [31]. SPHKs are involved in a multitude of disease processes, especially those of hyperproliferative and inflammatory nature, e.g., cancer, insulin resistance, and diabetic vascular complications [7,14,48]. Although limited data is available on the significance of the SPHK-S1P axis in normal and complicated pregnancies, studies are accumulating on their crucial roles, e.g., in decidualization, placentation, placental vascularization, and early pregnancy success [36]. Intriguingly, we show enhanced SPHK expression and activity in T2D placentae and increased placental SPHK levels in both types of diabetic pregnancies affected by fetal macrosomia. As the cytotrophoblast layer is the most metabolically active cell layer of the placenta [52] and S1P has been reported to downregulate cytotrophoblast differentiation into syncytiotrophoblasts, increased S1P signaling could aid in preserving the metabolic capacity of these cells, to meet the high oxidative phosphorylation demands and ensure active substrate transfer to the fetus [35]. In contrast, no changes in SPHK protein were found in pregnancies of women with pre-existing diabetes who develop preeclampsia, despite elevated levels of sphingoid-1-phosphates. In preeclampsia, a high turnover of ceramide substrates into sphingoid-1-phosphates could be a protective mechanism preventing the accumulation of pro-death lipid metabolites.

Despite the observed SPHK upregulation in both types of diabetic pregnancies with fetal macrosomia, increased sphingoid-1-phosphate levels were only recorded in T1DLGA placentae. This could result from the relatively low sample sizes available for these analyses. On the other hand, this finding is also in line with the high turnover rate and compartmentalization that characterize S1P synthesis [48]. Furthermore, in contrast to high S1P blood concentrations, S1P tissue concentrations are generally low, as much of the synthesized S1P is secreted [48] or broken down by S1P lyase, which is also an important regulator of S1P activity and tissue gradients [53,54]. In support of this notion, mouse studies suggest that SPHK overexpression may be coupled to enhanced S1P breakdown [41,55]. At least three out of the five G-protein coupled receptors that mediate S1P and Sa1P signaling have so far been identified in the placenta (S1PR1-S1PR3) [35], and the SPHK-S1P-S1PR axis has been implicated in a multitude of processes, from placental and embryonic development to trophoblast differentiation [56,57,58] and epigenetic modification [59]. Hence, the functions of both intracellular and excreted S1P in diabetic placentae and their potential significance in offspring health and development are an interesting avenue for future studies.

Our in vitro experiments suggest that exposure of placental cells to hyperglycemia and, to a lesser extent, hyperinsulinemia, may in part contribute to the upregulation of the SPHK-sphingoid-1-phosphate axis. Although data are scarce in trophoblast cells, increased SPHK activity has been previously reported in the aorta, heart [60], and kidney [61] of streptozotocin-induced diabetic rats, and high glucose conditions have been shown to upregulate SPHK1 in human umbilical vein endothelial cells [60] and glomerular mesangial cells [61]. Moreover, a study in breast cancer cells has shown that insulin can also mediate mitogenic effects via both SPHK1 and SPHK2 [29]. These studies are in accordance with our findings in placental JEG3 cells. Placental hypervascularization and surface area enlargement are typical of diabetic placentae, especially in association with fetal macrosomia [62,63], and evidence suggests that high fetal and maternal insulin levels could be among the stimuli leading to these morphological changes [63,64]. Hence, it is plausible that SPHK upregulation dominated T2D pregnancies and pregnancies associated with fetal macrosomia, i.e., pregnancies characterized by high fetal and/or maternal insulin levels in addition to hyperglycemia. It is possible that in diabetic pregnancies affected by processes that restrict fetal growth (poorer placental vascularization, hypertension, diabetic vasculopathy), downregulation of SPHK triggers opposing processes, in line with what has been previously demonstrated in IUGR [18].

A strength of our study is the comparably large total number of placental samples obtained from control and diabetic pregnancies. On the other hand, considering that diabetes is a heterogenous condition, a clear limitation is that the sample sizes per complication subgroup (preeclampsia, macrosomia) were smaller. All patients were diagnosed and followed up at the same hospital (MSH) according to uniform clinical guidelines, and standardized BioBank protocols were utilized in the collection of clinical specimens. For the lipidomic analyses, we utilized a highly sensitive and selective LC-MS/MS technique, which enabled even the measurement of lipid species with low tissue concentrations. In western blotting experiments, we used the currently recommended approach of total protein normalization with stain-free technology, which has been shown more reliable as compared to housekeeping proteins as loading controls [23]. Among the weaknesses of our study is the limited availability of information on maternal glycaemic control and no data on maternal blood lipid profile or other metabolic parameters, such as those reflecting insulin resistance. Due to the use of pre-collected biobank samples, we also lacked fetal (cord) blood samples, which could have been analyzed for markers reflecting fetal metabolism, and information on neonatal anthropometric variables, apart from relative birth weight. We also lacked information on placental histopathological and morphological changes.

## 5. Conclusions

In conclusion, our study provides novel evidence for distinct alterations of placental sphingolipid metabolism in T1D and T2D pregnancies. Collectively, our results suggest that the placental milieu in diabetes tilts the sphingolipid rheostat toward lower levels of ceramides, favoring the production of sphingoid-1-phosphates, especially in T2D (Graphical abstract). Furthermore, our results demonstrate that placental ceramide metabolism–with close linkages to central cellular signaling pathways such as those regulating cell fate and energy metabolism–is remarkably heterogenous in maternal pre-gestational diabetes, depending on the type of diabetes, associated complications, and clinical characteristics such as preeclampsia, fetal macrosomia, and poor glycemic control. Hence, changes in sphingolipid metabolism during pregnancy in a spectrum of diabetes may serve as footprints for predicting the onset of associated complications such as preeclampsia. This could be achieved by analyzing cargo of placental extracellular vesicles to identify changes in sphingolipids across gestation that could be used to predict diabetes-associated adverse outcomes. Considering the significance of in utero metabolic milieu in the programming of offspring metabolism and cardiovascular health, our results call for future studies addressing the role of these placental metabolic alterations in specific short- and long-term outcomes.

## Figures and Tables

**Figure 1 biomedicines-11-00932-f001:**
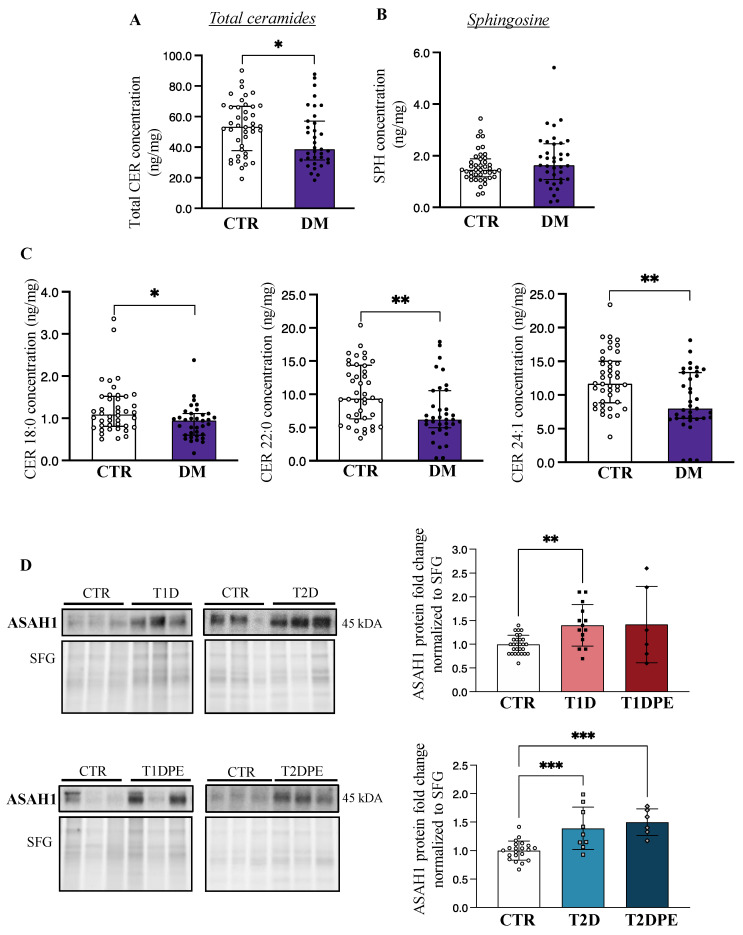
Placental ceramide and sphingosine content and acid ceramidase (ASAH1) levels in pre-gestational diabetes. (**A**) Placental total ceramide (CER) and (**B**) sphingosine (SPH) content in non-diabetic control pregnancies (CTR; *n* = 42) vs. type 1 and type 2 diabetes (DM; *n* = 35–37). (**C**) Placental CER 18:0, 22:0, and 24:1 levels CTR vs. DM. Values are median ± IQR. (**D**) Representative Western blots and associated densitometry analysis of placental ASAH1 in CTR (*n* = 28), type 1 diabetes (T1D; *n* = 14), T1D with preeclampsia (T1DPE; *n* = 6), type 2 diabetes (T2D; *n* = 9), and T2D with preeclampsia (T2DPE; *n* = 6). Data are mean ± SD. * *p* < 0.05, ** *p* < 0.01, *** *p* < 0.001. DM: Diabetes Mellitus.

**Figure 2 biomedicines-11-00932-f002:**
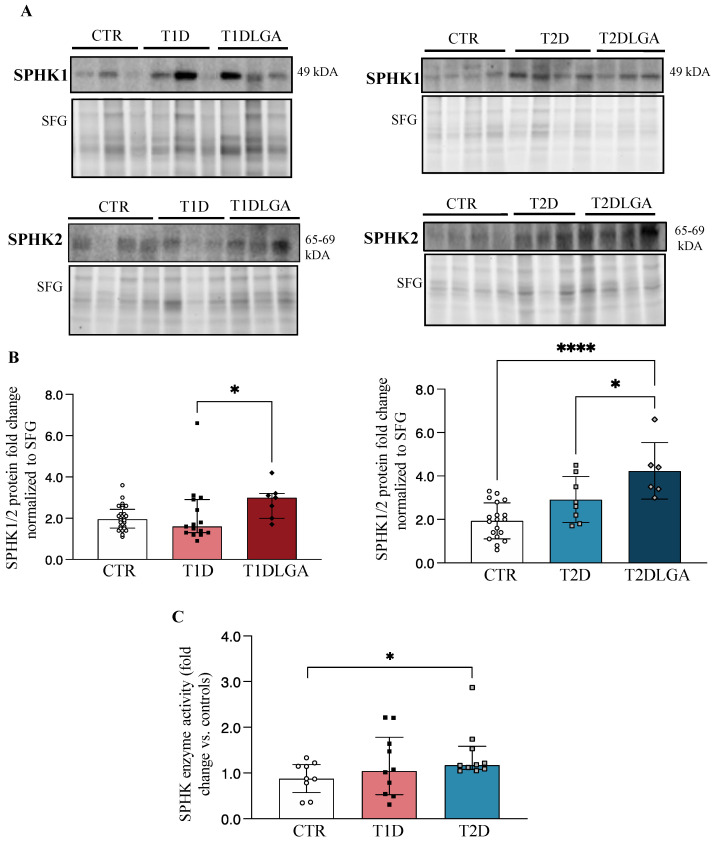
Sphingosine kinase (SPHK) levels and activity in diabetic placentae and in vitro conditions mimicking diabetes. ((**A**), left panel) Representative Western blots for SPHK1 and SPHK2 in placental lysates from type 1 diabetes (T1D) with and without fetal macrosomia (LGA) vs. Controls (CTR). ((**A**), right panel) Representative Western blots for SPHK1 and SPHK2 in placental lysates from type 2 diabetes (T2D) with and without fetal macrosomia (LGA) vs. Controls (CTR). ((**B**), left panel). Densitometric analyses of total SPHK (SPHK1+SPHK2) levels in CTR (*n* = 28), T1D (*n* = 15), and T1DLGA (*n* = 7) placentae (data are median ± IQR), and in ((**B**), right panel) CTR (*n* = 19), T2D (*n* = 8) and T2DLGA (*n* = 6) placentae (data are mean ± SD). (**C**) SPHK activity in CTR, T1D, and T2D pregnancies (data are median ± IQR). * *p* < 0.05; **** *p* < 0.0001.

**Figure 3 biomedicines-11-00932-f003:**
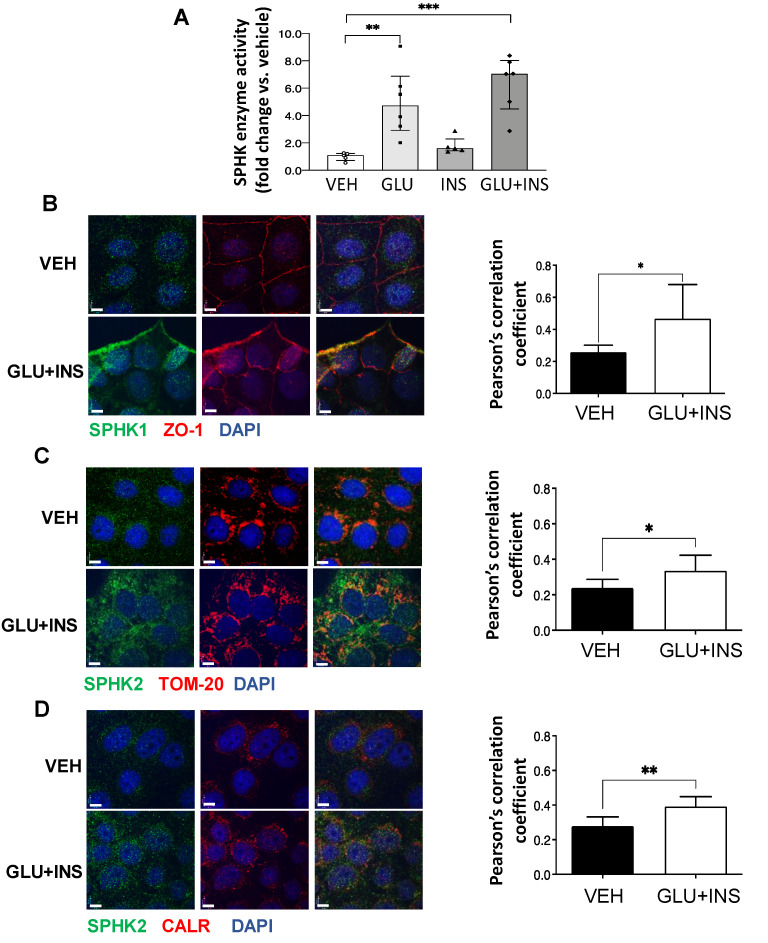
Subcellular distribution of sphingosine kinase (SPHK) isoenzymes in placental JEG3 cells in response to diabetic conditions. (**A**) SPHK activity in choriocarcinoma JEG-3 cells exposed to VEH (Eagle’s Minimal Essential Media), GLU (25 mM), INS (0.85 µM), or GLU+INS. Data are median ± IQR. * *p* < 0.05, ** *p* < 0.01, *** *p* < 0.001, (**B**–**D**) Immunofluorescence (IF) analysis of SPHK1 and 2 in choriocarcinoma JEG-3 cells exposed to VEH (Eagle’s Minimal Essential Media) or GLU (25 mM) + INS (0.85 µM). JEG3 were stained for (**B**) SPHK1 (green), nuclei (DAPI blue) and plasma membrane marker ZO-1 (red); (**C**) SPHK2 (green), nuclei (DAPI blue) and mitochondrial marker TOM20 (red); and (**D**) SPHK2 (green), nuclei (DAPI blue) and endoplasmic reticulum marker calreticulin (CRT) (red). Colocalization analysis (Pearson’s correlation coefficients) was performed on 7 images of VEH-treated cells and 8 images of GLU+INS-treated cells from 2 separate experiments. Data are mean ± SD. * *p* < 0.05; ** *p* < 0.01. Scale bars are equivalent to 7 µm.

**Table 1 biomedicines-11-00932-t001:** Maternal and perinatal characteristics of women with type 1 and type 2 diabetes and controls with non-diabetic pregnancies.

	Control (*n* = 49)	T1D (*n* = 28)	T2D (*n* = 21)	*p* Value
Maternal age (years)	33 (22–41) [42]	32 (20–43)	33 (25–50)	0.240
Nulliparous	16 (38.1) [42]	15 (53.6)	9 (42.9)	0.428
Age at diabetes diagnosis (years)	NA	15 (8–22) [23]	29 (15–37) [15]	<0.001
Diabetes duration (years)	NA	16 (10/22) [23]	3 (2/9) [15]	0.004
Maternal BMI (kg/m^2^)	21.9 (20.0/26.0) [39]	25.9 (23.8/30.9) [26]	32.3 (27.7/38.4) [20] †	<0.001
Gestational weight gain (GWG) (kg)	15.3 (5.3) [37]	15.0 (6.3) [26]	14.8 (9.3) [17]	0.963
GWG below IOM recommendations	1 (2.6) [39]	2 (7.7) [26]	0 (0) [16]	0.584
GWG above IOM recommendations	19 (48.7) [39]	17 (60.7) [28]	12 (70.6) [17]	0.288
SBP < 20 weeks’ gestation (mmHg)	113.0 (110.0/120.0) [18]	119.00 (110.0/130.0)	124.0 (120.0/131.50) [11]	0.084
DBP < 20 weeks’ gestation (mmHg)	68.0 (60.0/75.0) [18]	71.0 (70.0/76.0)	76.0 (69.50/80.0) [11]	0.197
SBP ≥ 20 weeks’ gestation (mmHg)	111.0 (122.0/132.0) [41]	136.0 (129.50/151.50) †	149 (138.0/170.0) †	<0.001
DBP ≥ 20 weeks’ gestation (mmHg)	74.0 (70.0/82.0) [41]	85.0 (80.50/97.50) †	91.0 (81.0/101.0)†	<0.001
Diabetic retinopathy	NA	1 (3.6)	3 (14.3)	0.301
Diabetic nephropathy	NA	1 (3.6)	1 (4.8)	1.000
Chronic hypertension	0 (0)	4 (14.3) *	5 (23.8) *	<0.001
Gestational hypertension	3 (6.1)	2 (7.1)	6 (28.6) ‡	0.027
Preeclampsia	0 (0)	6 (21.4) ‡	6 (28.6) ‡	<0.001
HbA_1c_ (mmol/mol, %)				
Pre-pregnancy	NA	8.5 (8.3/8.8) [3]	8.7 (8.6/8.8) [2]	0.564
First value in first trimester	NA	7.4 (6.9/8.1) [7]	8.6 (7.9/8.8) [3]	0.360
Highest value in second trimester	NA	6.2 (6.0/6.5) [9]	6.7 (6.4/7.7) [8]	0.135
Last value in third trimester	NA	6.7 (6.1/7.1) [20]	6.2 (6.1/6.9) [10]	0.401
Delivery mode				0.352
Vaginal delivery	7 (16.7) [42]	8 (28.6)	4 (19.0)	
Cesarean section without labor	31 (73.8) [42]	14 (50.0)	14 (66.7)	
Cesarean section after labor	4 (9.5) [42]	6 (21.4)	3 (14.3)	
Gestational age at delivery (weeks)	39.0 (38.7/39.6) [43]	37.6 (36.9/38.3) †	37.9 (35.3/38.6) †	<0.001
Fetal sex, male/female	24 (57.1)/18 (42.9)	14 (50.0)/14 (50.0)	7 (33.3)/14 (66.7)	0.227
Birth weight (g)	3379.3 (355.4) [46]	3518.6 (691.7)	3133.3 (1264.0)	0.201
Relative birth weight (SD units)	0.09 (0.85) [42]	1.19 (1.41) ‡	0.93 (1.82) *	0.002
Fetal macrosomia (relative birth weight > +2.0 SD units)	0 (0) [42]	7 (25.0)	6 (28.6)	<0.001
Placental weight (g)	627.5 (530.0/710.0) [42]	657.5 (570.0/800.0) [22]	690.0 (475.0/842.0)	0.520
Apgar score at 5 min	9 (8–9) [41]	9 (8–9) [27]	9 (7–9) [20]	0.390

Values for continuous variables are mean (SD) or median (25th/75th percentile or range), and for categorical variables frequencies (%). Number of subjects is shown in square brackets if different. *p* values for comparisons of three groups are shown in the right column (*p* value) and those for pairwise post-hoc comparisons are indicated by symbols. * *p* < 0.05 Controls vs. T1D/T2D; ‡ *p* < 0.01 Controls vs. T1D/T2D; † *p* < 0.001 for Controls vs. T1D/T2D. Type 1 diabetes (T1D); Type 2 diabetes (T2D); Institutes of Medicine (IOI); Not applicable (NA); Systolic blood pressure (SBP); Diastolic blood pressure (DBP).

**Table 2 biomedicines-11-00932-t002:** Placental sphingolipids in maternal pre-existing diabetes with and without preeclampsia.

Ceramides	CTR (*n* = 42)	T1D (*n* = 20)	T1DPE (*n* = 6)	*p* Value
CER 14:0	0.12 (0.09/0.16)	0.15 (0.11/0.25)	0.11 (0.08/0.12)	0.057
CER 16:0	7.32 (5.44/9.24)	7.22 (5.94/8.67)	9.73 (8.52/11.02)	0.052
CER 18:0	1.09 (0.82/1.52)	0.94 (0.64/1.02)	1.24 (0.87/1.40)	0.040
CER 20:0	0.59 (0.48/0.86)	0.50 (0.44/0.71)	0.69 (0.61/0.74)	0.108
CER 22:0	9.36 (6.42/14.32)	6.32 (4.34/9.38)	14.42 (8.40/16.08) ‡	0.016
CER 24:0	21.00 (14.88/26.60)	15.06 (10.69/25.30)	30.00 (29.80/31.60) ‡	0.022
CER 24:1	11.67 (8.94/14.88)	8.17 (6.47/13.62) *	10.65 (8.72/11.76)	0.033
Total CER	53.34 (33.36/66.57)	38.72 (31.67/51.81)	66.99 (58.87/71.28) ‡	0.022
**Sphingoid-1-phosphates**	**CTR** (*n* = 18)	**T1D** (*n* = 18)	**T1D with PE** (*n* = 6)	
S1P+Sa1P	0.093 (0.067/0.138)	0.087 (0.079/0.125)	0.177 (0.167/0.361) †	0.035
**Ceramides**	**CTR** (*n* = 42)	**T2D** (*n = 15)*	**T2DPE** (*n* = 6)	*p* value
CER 14:0	0.12 (0.09/0.16)	0.11 (0.10/0.21)	0.08 (0.07/0.14)	0.683
CER 16:0	7.32 (5.44/9.24)	8.04 (6.20/10.63)	7.09 (4.84/11.80)	0.660
CER 18:0	1.09 (0.82/1.52)	0.86 (0.59/1.16)	0.62 (0.55/0.74)†	0.019
CER 20:0	0.59 (0.48/0.86)	0.61 (0.46/0.71)	0.45 (0.42/0.63)	0.301
CER 22:0	9.36 (6.42/14.32)	6.18 (5.48/8.74)	6.62 (3.86/9.76)	0.090
CER 24:0	21.00 (14.88/26.60)	17.84 (12.77/23.70)	11.80 (5.86/19.16) †	0.040
CER 24:1	11.67 (8.94/14.88)	7.78 (6.62/12.59) *	5.47 (3.26/8.04) †	0.001
Total CER	53.34 (33.36/66.57)	38.63 (31.89/58.72)	35.60 (18.90/45.43)	0.070
**Sphingoid-1-phosphates**	**CTR** (*n* = 18)	**T2D** (*n =* 12)	**T2D with PE** (*n* = 4)	*p* value
S1P+Sa1P	0.093 (0.067/0.138)	0.112 (0.054/0.282)	0.457 (0.231/0.891)	0.188

Lipid concentrations (ng/mg of tissue). Values are median (interquartile range). Ceramide species and bioactive sphingoid-1-phosphates concentrations in control (CTR) vs. type 1 diabetic (T1D) placentae with and without PE (*upper panel*); and in control vs. type 2 diabetic (T2D) placentae with and without PE (*lower panel*). *p* values for comparisons of three groups are shown in the right-most column and those for pairwise post-hoc comparisons are indicated by following symbols: * *p* < 0.05 CTR vs. T1D/T2D; † *p* < 0.05 CTR vs. T1DPE/T2DPE; ‡ *p* < 0.05 T1D/T2D vs. T1DPE/T2DPE. Ceramides (CER); Sphingosine-1-phosphate (S1P); sphinganine-1-phosphate (Sa1P).

**Table 3 biomedicines-11-00932-t003:** Placental sphingolipids in maternal pre-existing diabetes with and without fetal macrosomia.

Ceramides	CTR (*n* = 42)	T1D (*n* = 13)	T1DLGA (*n* = 7)	*p* Value
CER 14:0	0.12 (0.09/0.16)	0.15 (0.08/0.17)	0.35 (0.18/0.71) †	0.013
CER 16:0	7.32 (5.44/9.24)	6.48 (5.56/7.06)	8.62 (8.04/10.31) ‡	0.022
CER 18:0	1.09 (0.82/1.52)	0.82 (0.65/1.01)	0.97 (0.73/1.04)	0.060
CER 20:0	0.59 (0.48/0.86)	0.52 (0.45/0.64)	0.49 (0.44/0.64)	0.211
CER 22:0	9.36 (6.42/14.32)	7.05 (5.58/10.56)	4.02 (2.46/7.06)	0.035
CER 24:0	21.00 (14.88/26.60)	15.70 (10.48/28.00)	14.24 (11.74/25.30)	0.275
CER 24:1	11.67 (8.94/14.88)	8.36 (6.52/10.92) *	7.98 (6.88/13.62)	0.037
Total CER	53.34 (33.36/66.57)	39.62 (35.83/49.55)	37.78 (31.57/60.14)	0.116
**Sphingoid-1-phosphates**	**CTR** (*n* = 18)	**T1D** (*n* = 11)	**T1D with LGA** (*n* = 7)	*p* value
S1P+Sa1P	0.093 (0.067/0.138)	0.085 (0.070/0.089)	0.220 (0.102/0.403) ‡	0.048
**Ceramides**	**CTR** (*n* = 42)	**T2D** (*n* = 9)	**T2DLGA** (*n* = 7)	*p* value
CER 14:0	0.12 (0.09/0.16)	0.13 (0.10/0.27)	0.10 (0.08/0.19)	0.498
CER 16:0	7.32 (5.44/9.24)	9.32 (6.08/11.92)	7.64 (6.86/8.32)	0.661
CER 18:0	1.09 (0.82/1.52)	0.81 (0.58/1.11)	0.99 (0.68/1.20)	0.222
CER 20:0	0.59 (0.48/0.86)	0.59 (0.38/0.68)	0.64 (0.54/0.73)	0.860
CER 22:0	9.36 (6.42/14.32)	6.03 (5.49/10.66)	6.49 (5.48/6.82)	0.193
CER 24:0	21.00 (14.88/26.60)	13.70 (13.02/25.40)	19.62 (12.52/22.00)	0.845
CER 24:1	11.67 (8.94/14.88)	7.78 (6.56/13.32)	8.58 (6.68/12.36)	0.040
Total CER	53.34 (33.36/66.57)	33.41 (32.13/57.07)	43.59 (31.63/60.36)	0.507
**Sphingoid-1-phosphates**	**CTR** (*n* = 18)	**T2D** (*n =* 8)	**T2D with LGA** (*n* = 6)	*p* value
S1P+Sa1P	0.093 (0.067/0.138)	0.162 (0.050/0.274)	0.106 (0.059/0.422)	0.727

Lipid concentrations (ng/mg of tissue). Values are median (interquartile range). Ceramide species and bioactive sphingoid-1-phosphates concentrations in control vs. type 1 diabetic (T1D) placentae with and without fetal macrosomia (LGA) (*upper panel)*; and in control vs. type 2 diabetic (T2D) placentae with and without LGA (*lower panel*). *p* values for comparisons of three groups are shown in the right column and those for pairwise post-hoc comparisons are indicated by following symbols: * *p* < 0.05 CTR vs. T1D; † *p* < 0.05 CTR vs. T1DLGA; ‡ *p* < 0.05 T1D vs. T1DLGA. Ceramides (CER); Total sphingoid-1-phosphates (Total sphingoid-1-p); Sphingosine-1-phosphate (S1P); sphinganine-1-phosphate (Sa1P).

**Table 4 biomedicines-11-00932-t004:** Placental concentrations of ceramides and lysosphingolipids in pregnancies complicated by T1D or T2D categorized into those with good or poor glycemic control according to HbA_1c_ concentration before delivery.

Ceramides	HbA_1c_ < 64 mmol/L (*n* = 25)	HbA_1c_ ≥ 64 mmol/L (*n* = 6)	*p* Value
CER 14:0	0.14 (0.09/0.22)	0.10 (0.10/0.12)	0.448
CER 16:0	6.90 (5.80/8.54)	9.82 (8.50/12.34)	0.010
CER 18:0	0.82 (0.65/0.99)	1.20 (1.11/1.26)	0.015
CER 20:0	0.54 (0.43/0.68)	0.58 (0.44/0.70)	0.865
CER 22:0	5.88 (4.66/10.56)	6.63 (5.24/8.40)	0.715
CER 24:0	13.08 (10.90/18.56)	28.20 (21.40/31.80)	0.009
CER 24:1	7.44 (6.52/10.92)	10.35 (8.72/13.44)	0.075
*Total CER*	35.93 (29.81/46.42)	55.86 (48.55/71.28)	0.012
**Sphingoid-1-phosphates**	**HbA_1c_ < 64 mmol/L**	**HbA_1c_ ≥ 64 mmol/L**	*p* value
S1P+Sa1P	0.094 (0.084/0.302)	0.140 (0.101/0.220)	0.511

Lipid concentrations (ng/mg of tissue). Values are median (interquartile range). Type 1 diabetes (T1D), Type 2 diabetes (T2D); Controls (CTR); Ceramides (CER); Sphingosine-1-phosphate (S1P); sphinganine-1-phosphate (Sa1P).

## Data Availability

Data sharing is not applicable to this article.

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
