# Peer review of "Distinct Changes in Placental Ceramide Metabolism Characterize Type 1 and 2 Diabetic Pregnancies with Fetal Macrosomia or Preeclampsia"

_biomedicines, 2023, doi:10.3390/biomedicines11030932_

Round 1

Reviewer 1 Report

In this manuscript, the authors examined the placental ceramide and sphingoid-1-phosphate content and metabolism in women with T1D and T2D. The authors also investigated whether common complications that burden mothers with diabetes (preeclampsia) and their fetuses (macrosomia/large-for-gestational age), are characterized by distinct placental sphingolipid profiles that could also shed light on the placental metabolic derangements implicated in the pathogenesis of these complex outcomes. The manuscript is overall clear and well written. However, I have some minor comments for the authors to revise.

  1. Scale bars should be added in Figure 3B, instead of using 63x.
  2. Limitations of this study should be discussed in the Discussion section.
  3. Earlier publications which support or contradict this study should be compared in the Discussion section.
  4. A illustration to show the potential mechanism of placental ceramide metabolism is highly recommended.

Reviewer 2 Report

Klemetti et al. They present an interesting, well-written manuscript with data of interest and great repercussion. The manuscript presents an adequate state of the art, with a correct methodology and adequately justified in the nature of the study. The results are innovative and are adequately written and described. Authors should improve Figure 1. Authors can make color histograms. Authors should make more self-explanatory figures.

The discussion is adequate, however, the authors should place more emphasis on the more translational aspects and clinical management.

The results support the conclusions of the manuscript.
